# Chinese natural compound decreases pacemaking of rabbit cardiac sinoatrial cells by targeting second messenger regulation of f-channels

Chiara Piantoni[1†‡], Manuel Paina[1†§], David Molla[1], Sheng Liu[2], Giorgia Bertoli[1], Hongmei Jiang[2], Yanyan Wang[3], Yi Wang[4], Yi Wang[5], Dario DiFrancesco[1], Andrea Barbuti[1], Annalisa Bucchi[1*], Mirko Baruscotti[1*]

[1]Department of Biosciences, The Cell Physiology Lab and "Centro Interuniversitario di Medicina Molecolare e Biofisica Applicata", Università degli Studi di Milano, Milano, Italy; [2]Department of Physiology and Pathophysiology, School of Basic Medical Science, Tianjin Medical University, Tianjin, China; [3]School of Integrative Medicine, Tianjin University of Traditional Chinese Medicine, Tianjin, China; [4]College of Pharmaceutical Sciences, Zhejiang University, Hangzhou, China; [5]Institute of Traditional Chinese Medicine Tianjin University of Traditional Chinese Medicine, Tianjin, China

*For correspondence:
annalisa.bucchi@unimi.it (AB);
mirko.baruscotti@unimi.it (MB)

[†] These authors contributed equally to the paper

Present address: [‡]Institute of Neurophysiology, Hannover Medical School, Hannover, Germany; [§]Axxam S.p.A, Bresso, Italy

**Abstract** Tongmai Yangxin (TMYX) is a complex compound of the Traditional Chinese Medicine (TCM) used to treat several cardiac rhythm disorders; however, no information regarding its mechanism of action is available. In this study we provide a detailed characterization of the effects of TMYX on the electrical activity of pacemaker cells and unravel its mechanism of action. Single-cell electrophysiology revealed that TMYX elicits a reversible and dose-dependent (2/6 mg/ml) slowing of spontaneous action potentials rate (−20.8/−50.2%) by a selective reduction of the diastolic phase (−50.1/−76.0%). This action is mediated by a negative shift of the $I_f$ activation curve (−6.7/−11.9 mV) and is caused by a reduction of the cyclic adenosine monophosphate (cAMP)-induced stimulation of pacemaker channels. We provide evidence that TMYX acts by directly antagonizing the cAMP-induced allosteric modulation of the pacemaker channels. Noticeably, this mechanism functionally resembles the pharmacological actions of muscarinic stimulation or β-blockers, but it does not require generalized changes in cytoplasmic cAMP levels thus ensuring a selective action on rate. In agreement with a competitive inhibition mechanism, TMYX exerts its maximal antagonistic action at submaximal cAMP concentrations and then progressively becomes less effective thus ensuring a full contribution of $I_f$ to pacemaker rate during high metabolic demand and sympathetic stimulation.

## Editor's evaluation

Tongmai Yangxin (TMYX) is a complex compound of Traditional Chinese Medicine used to treat several cardiac rhythm disorders; however, no information regarding its mechanism of action is available. This study provides mechanistic insight into where TMYX acts to inhibit the pacemaking current called If. In some respects TMYX behaves like a β blocker or muscarinic antagonist, but it works to inhibit the ion channel to maximum effect when cAMP concentrations are low, thus allowing the full effect of sympathetic stimulation to still occur on $I_f$ when metabolic rate is high. This compound therefore has the potential for high therapeutic utility to control cardiac arrhythmia.

## Introduction

Traditional Chinese Medicine (TCM), one of the oldest organized healing systems in human history, is based on a holistic view of both the disease state and the associated therapy. For this reason, the perfect synthesis of TCM pharmacology is based on complex drugs composed of a mixture of different elements/herbs whose aim is to target the causes of the disease, modulate other aspects of the body wellness, and contrast toxicity (*Chen et al., 2006*). Western medicine follows a different perspective since it focuses on the molecular mechanisms and defines this level as its therapeutic target. Despite these differences, both approaches have reached reliable standards (*Tang and Huang, 2013*; *Tu, 2016*). These different views are now converging thanks to modern pharmacological and molecular studies whose approach is to experimentally challenge the efficacy of TCM drugs and to isolate active molecules that could represent novel acquisitions to the western pharmacopeia (*Burashnikov et al., 2012*; *Chen et al., 2006*; *Efferth et al., 2007*; *Lao et al., 2009*; *Tang and Huang, 2013*).

Based on these premises, we focused our study on Tongmai Yangxin (TMYX), a TCM botanical drug composed by at least 80 single molecular components among which flavonoids, coumarins, iridoid glycosides, saponins, and lignans (*Tao et al., 2015*). In China, TMYX is used (4–8 g qd, *Fan et al., 2021*) to treat several diseases including cardiovascular conditions (such as CAD, palpitation, heart failure, and angina). Interestingly, metabolomics analysis carried out in a registered clinical trial on stable angina patients has highlighted a reduction of serum markers of cardiac metabolic disorders, oxidative stress, and inflammation (*Cai et al., 2018*; *Fan et al., 2016*; *Guo et al., 2020*). In addition to this cardio-protective role, TMYX is also used as an antiarrhythmic agent (*Cai et al., 2018*; *National Pharmacopeia Committee, 2015*), but the mechanisms underlying this action are unknown. We therefore carried out *in-vitro* investigations in rabbit sinoatrial node (SAN) cells to verify the hypothesis that TMYX is also able to modulate their spontaneous activity. Our data confirm that TMYX reduces the rate of SAN cells by selectively reducing the slope of the diastolic depolarization. This action is similar to that of ivabradine which is a selective blocker of the pacemaker current ($I_f$) and, at present, the only pure heart rate-lowering agent approved for clinical use in several western countries (*Koruth et al., 2017*). We have further discovered that TMYX modulates the whole-cell $I_f$ current by inducing a cholinergic-like shift of the voltage dependence of channels activation, and the underlying mechanism is a competitive antagonism of the cyclic adenosine monophosphate (cAMP)-induced channel activation. Since the $I_f$ current is a major contributor of the early part of the diastolic depolarization phase of pacemaker cells, its selective block is associated with bradycardia without negative inotropic effects (typical, for example, of β-blocker agents). Despite TMYX and ivabradine inhibit the $I_f$ current with different molecular mechanisms, both these mechanisms functionally converge to a selective modulation of the early part of the diastolic depolarization of SAN cells and this raises a potential pharmacological interest in the active principle of this TCM drug.

## Results

We first investigated whether TMYX could modify the spontaneous electrical activity of rabbit SAN myocytes. In *Figure 1A* representative time-courses of action potential (AP) rate (top) and sample AP traces (bottom), recorded in the absence (control) and in the presence of two different concentrations (2 and 6 mg/ml) of TMYX, are shown. TMYX caused a reversible and dose-dependent rate slowing (2 mg/ml: –20.8 ± 1.6%, n = 12 and 6 mg/ml: –50.2 ± 6.5%, n = 8 from mean control values of 3.6 ± 0.1 and 3.9 ± 0.3 Hz, respectively). The analysis was then extended over a wider range of concentrations and the Hill fitting of the experimental dose-response data points distribution yielded a half-inhibitory value (k) of 4.9 mg/ml, a maximal block value ($y_{max}$) of 92.7%, and a Hill coefficient (h) of 1.3 (*Figure 1B*). At the highest concentration tested (60 mg/ml), the average rate reduction was 86.7 ± 6.3% (n = 6); in three of these cells the activity was completely abolished. In all experiments, the effect of TMYX on rate was fully reversible after washout.

To dissect the action of the drug during the various phases of the AP, we quantitatively evaluated specific AP parameters (early diastolic depolarization [EDD]; AP duration [APD]; maximum diastolic potential [MDP]; take-off potential [TOP]) in the absence and during perfusion of different doses of TMYX (0.2, 0.6, 2, and 6 mg/ml). As shown in *Figure 1B*, the spontaneous rate was significantly reduced at all doses investigated, and this effect was for the largest part caused by a significant decrease of the EDD (rate: –6.3 ± 1.2%, –8.2 ± 0.9%, –20.8 ± 1.6%, –50.2 ± 6.5%; EDD:–9.7 ± 1.5%,

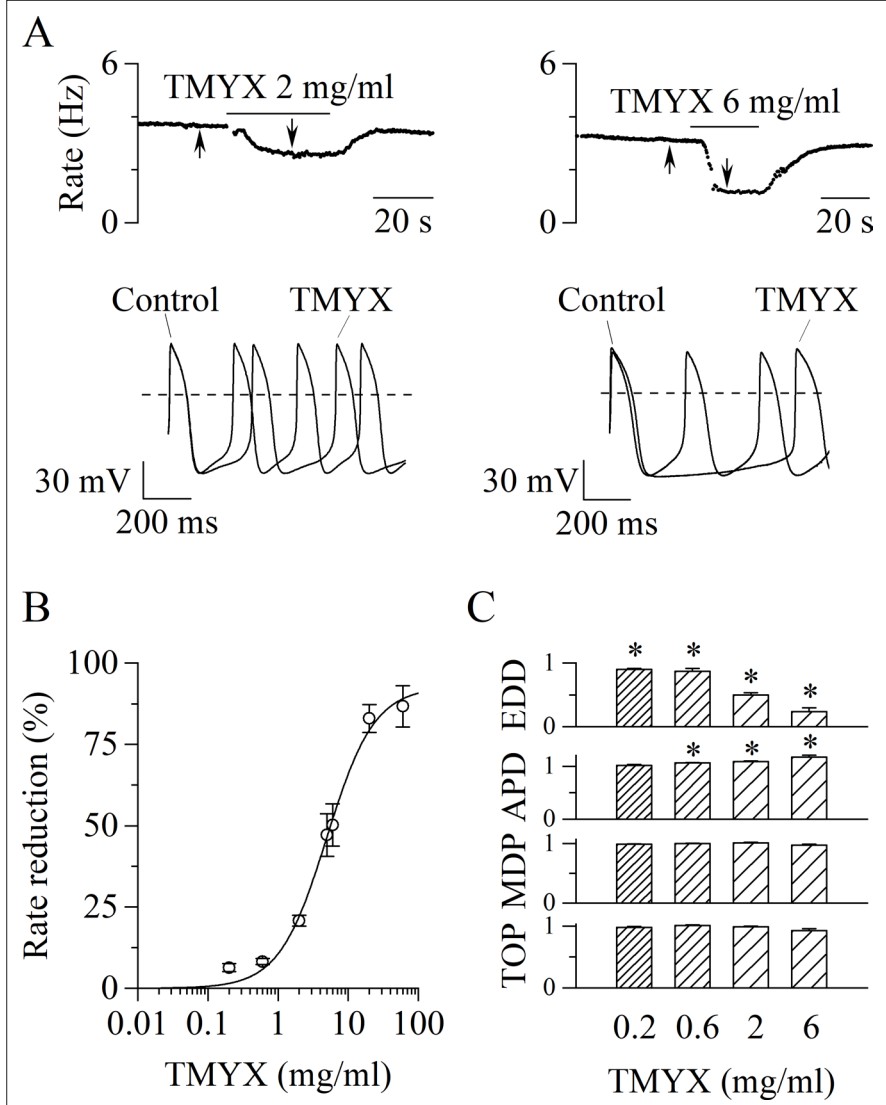

**Figure 1.** Tongmai Yangxin (TMYX) reduces the spontaneous rate of rabbit sinoatrial node (SAN) myocytes. (**A**) Representative time-courses (top) and sample traces (bottom) of spontaneous action potentials (APs) recorded from rabbit SAN cells in control conditions and in the presence of TMYX (2 and 6 mg/ml). Here and in other figures the arrows indicate the time of recording of the sample traces. (**B**) Dose-response relationship of the AP rate reduction induced by TMYX; each point represents the mean ± SEM% value obtained at the following doses: 0.2, 0.6, 2, 5, 6, 20, 60 mg/ml (n = 68). The Hill fitting (full line, $y = y_{max}/(1+(k/x)^h)$) yielded the following values: $y_{max}$ = 92.7%, k = 4.9 mg/ml, and h = 1.3. (**C**) Summary of the effects of TMYX on the AP parameters (n = 7–12, details in the Materials and methods) normalized to the corresponding control values. Statistical analysis was carried out prior to normalization, *p < 0.01 vs. control (Student's paired t-test). Data related to this figure are available in *Figure 1—source data 1*.

The online version of this article includes the following source data for figure 1:

**Source data 1.** Quantification and statistics of TMYX effect on spontaneous activity (APs) recorded from single sinoatrial node cells.

---

−12.8 ± 4.4%, −50.1 ± 3.7%, −76.0 ± 5.7%). A small increase of the APD was observed at doses ≥ 0.6 mg/ml (6.2 ± 0.8%, 8.7 ± 1.1%, 17.1 ± 3.7%); TOP and MDP were not affected (*Figure 1C*).

*Figure 1* provides evidence that TMYX lowers AP rate mainly by affecting the pacemaker mechanisms governing the EDD process. Since the $I_f$ current is relevant to the generation of this phase (*Bucchi et al., 2007*; *DiFrancesco, 1993*), we wondered whether this current could be a target of TMYX. We initially explored the effects of TMYX both on the voltage dependence and on the maximal

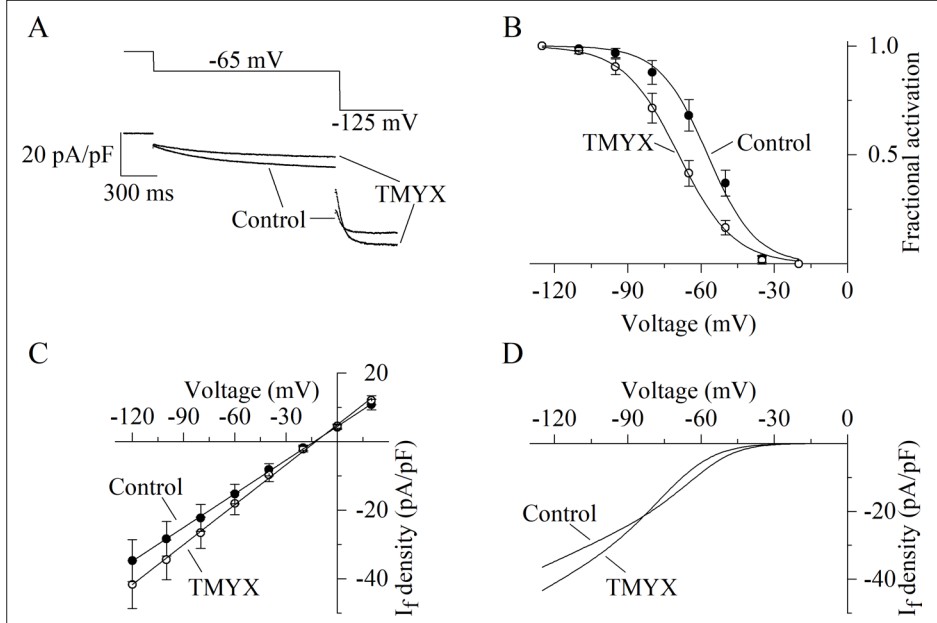

**Figure 2.** Dual action of Tongmai Yangxin (TMYX) on the voltage dependence and maximal conductance of the $I_f$ current. (**A**) Representative whole-cell currents elicited by a double-step protocol (–65 mV/1.5 s and –125 mV/0.5 s; holding potential –35 mV) before (control) and during drug (TMYX 6 mg/ml) perfusion. (**B**) Voltage-dependent activation curves obtained in control conditions (filled circles) and during TMYX perfusion (empty circles). Boltzmann fitting (full lines, y = 1/(1 + exp((V−V½)/s))) of mean fractional activation values (n = 7 cells) yielded the following half-activation (V½) and inverse-slope factors (s) parameters: –57.3 and 9.8 mV (control) and –69.2 and 11.3 mV (TMYX); the shift caused by TMYX is statistically significant (p < 0.01, extra sum-of-squares F test). (**C**) Mean fully activated current/voltage (I/V) relations measured before (filled circles) and during drug perfusion (empty circles, n = 5 cells). Linear fitting yielded reversal potentials of –13.6 and –12.7 mV and slopes of 0.328 and 0.389 (pA/pF)/mV in control and in the presence of TMYX, respectively; the slopes are significantly different (p < 0.01, linear regression analysis test). (**D**) Steady-state I/V fitting curves obtained by multiplying the activation curves (Boltzmann fitting, panel B) and fully activated I/V relation (linear fitting, panel C) in control condition and in the presence of the drug. Data related to this figure are available in *Figure 2—source data 1*.

The online version of this article includes the following source data and figure supplement(s) for figure 2:

**Source data 1.** Quantification and statistics of TMYX (6 mg/ml) effect on the voltage-dependence and fully-activated I/V properties of the funny current.

**Figure supplement 1.** Dual action of Tongmai Yangxin (TMYX) (2 mg/ml) on the voltage dependence and maximal conductance of the $I_f$ current.

**Figure supplement 1—source data 1.** Quantification and statistics of TMYX (2 mg/ml) effect on the voltage-dependence and fully-activated I/V properties of the funny current.

**Figure supplement 2.** Effects of Tongmai Yangxin (TMYX) (6 mg/ml) on the voltage dependence of the steady-state $I_f$ current measured at different voltages.

**Figure supplement 2—source data 1.** Quantification of TMYX (6 mg/ml) effect on the voltage-dependence of the steady-state funny current.

conductance of $I_f$. To this aim we used a double-pulse protocol which allows to observe the effect of a drug on the current both near the half-activation voltage (–65 mV) and at full activation (–125 mV, *Figure 2A*). Perfusion of SAN cells with TMYX 6 mg/ml modified the current at both voltages, but in opposite directions: in the sample recordings shown in *Figure 2A*, at –65 mV the current was reduced by 40.5%, while at –125 mV was increased by 12.9%. This apparently paradoxical behavior was observed in all cells investigated (n = 6 cells) and a possible explanation requires the combination of two contrasting effects: a negative shift of the activation curve and an increase of the maximal conductance. To evaluate this possibility, we carried out a quantitative characterization of these effects. The activation curves of the $I_f$ current were measured in n = 7 cells before and during TMYX (6 mg/ml) and mean ± SEM values are plotted in *Figure 2B*. Boltzmann fitting of experimental data

confirmed a significant hyperpolarizing shift of the activation curve (11.9 mV, p < 0.01). If considered alone this effect would tend to decrease the contribution of the current to pacemaker depolarization, hence to rate slowing.

To better investigate the TMYX-induced current increase at –125 mV, we measured the fully activated current/voltage (I/V) relation in control condition and in the presence of TMYX (6 mg/ml, n = 5; *Figure 2C*). Linear fitting of mean ± SEM data confirmed that TMYX increased the slope of the fully activated I/V relation by 18.6%.

Data in *Figure 2A, B, and C* thus indicate that TMYX exerts functionally opposite effects on the voltage-dependent availability of the current, which is decreased, and on the maximal conductance, which is increased. This observation is better illustrated by considering the steady-state $I_f$ current curves (*Figure 2D*): at voltages more positive than the cross-over point (–83 mV) the prevalent effect of TMYX is a current reduction due to the leftward shift of its activation curve, while at more negative voltages the increase in conductance prevails. A similar effect was observed in the presence of a lower dose (2 mg/ml) of TMYX (*Figure 2—figure supplement 1*). To further strengthen this finding, a train of different activating steps was delivered prior to and during TMYX (6 mg/ml) exposure, and the mean ± SEM steady-state current amplitudes (n = 7 cells) are plotted in *Figure 2—figure supplement 2*. Fitting of experimental data with the following equation $I_{density}=(a*V + b)*(1/(1+ exp((V- V_{½})/s)))$, which combines the linear I/V behavior with the Boltzmann sigmoidal voltage dependence, confirmed the presence of a cross-over phenomenon in the steady-state I/V relations.

Taken together, data presented in *Figure 2*, *Figure 2—figure supplement 1*, *Figure 2—figure supplement 2* reveal that at diastolic voltages TMYX reduces the $I_f$ contribution by shifting its voltage dependence to more negative values. Since the cholinergic control of the $I_f$ current, and thus of SAN rate, operates via a similar mechanism, we asked whether one or more components of TMYX could act as muscarinic agonist. To address this point, we compared the effects of TMYX (6 mg/ml) and acetylcholine (ACh, 1 µM) on $I_f$ and on cell rate, as measured in the absence and presence of the muscarinic blocker atropine (10 µM).

If current traces recorded at –65 mV in control and in the presence of either TMYX (top) or ACh (bottom), delivered alone (left) or in combination with atropine (right), are shown in *Figure 3A*.

The TMYX-induced reduction of the $I_f$ current was not modified by atropine (TMYX: –34.6 ± 4.9%, TMYX + atropine: –33.5 ± 3.4%, n = 7), while atropine abolished the effect of ACh (n = 6; *Figure 3A and B*). In line with the findings on $I_f$, we also observed that the muscarinic block did not antagonize the rate-slowing effect elicited by TMYX on SAN cells (TMYX: –42.6 ± 5.8%; TMYX + atropine: –37.6 ± 4.8%, n = 6; *Figure 3C and D*). On the other hand, atropine abolished the ACh-induced rate slowing (n = 7; *Figure 3C and D*).

We also asked whether TMYX could exert its inhibitory action by interfering with adenosine, a well-known modulator of $I_f$ whose action is based on a negative shift of the activation curve (*Zaza et al., 1996*). Data shown in *Figure 3—figure supplement 1* also demonstrate that the adenosine receptor is not involved in the TMYX-induced inhibition of the current.

Although the role of the cAMP-dependent protein kinase (PKA)-induced phosphorylation of pacemaker channels is still an open issue (*Mika and Fischmeister, 2021*), a relevant study by *Liao et al., 2010* shows that in mice SAN cells PKA modulates the voltage dependence of $I_f$. We thus tested whether the effect of TMYX (2 mg/ml) on basal cell rate was dependent upon PKA activity. Data presented in *Figure 3—figure supplement 2* demonstrate that TMYX-induced reduction of basal cell rate was not affected by the presence of the PKA inhibitor H-89; indeed, mean reductions measured in the absence and in the presence of H-89 were –27.5% and –27.6%, respectively (n = 7).

After excluding the involvement of a direct activation of the muscarinic (and adenosine) receptors, and an inhibition of PKA we next asked whether TMYX could exert its effect on the $I_f$ activation curve by interfering with a downstream effector, and particularly on the cAMP-dependent modulation of the channel. This hypothesis was based on the well-established evidence that the voltage-dependent availability of $I_f$ is controlled by the binding/unbinding of cAMP molecules to the pacemaker f-channels (*DiFrancesco and Tortora, 1991*; *James and Zagotta, 2018*).

To investigate the existence of a possible functional interference between cAMP and TMYX, we performed the experiments shown in *Figure 4*, where the effect of TMYX (6 mg/ml) on the whole-cell $I_f$, measured in the diastolic range of potentials, was assessed in the absence (basal) and in the presence of two concentrations of cAMP in the pipette solution (10, 100 µM).

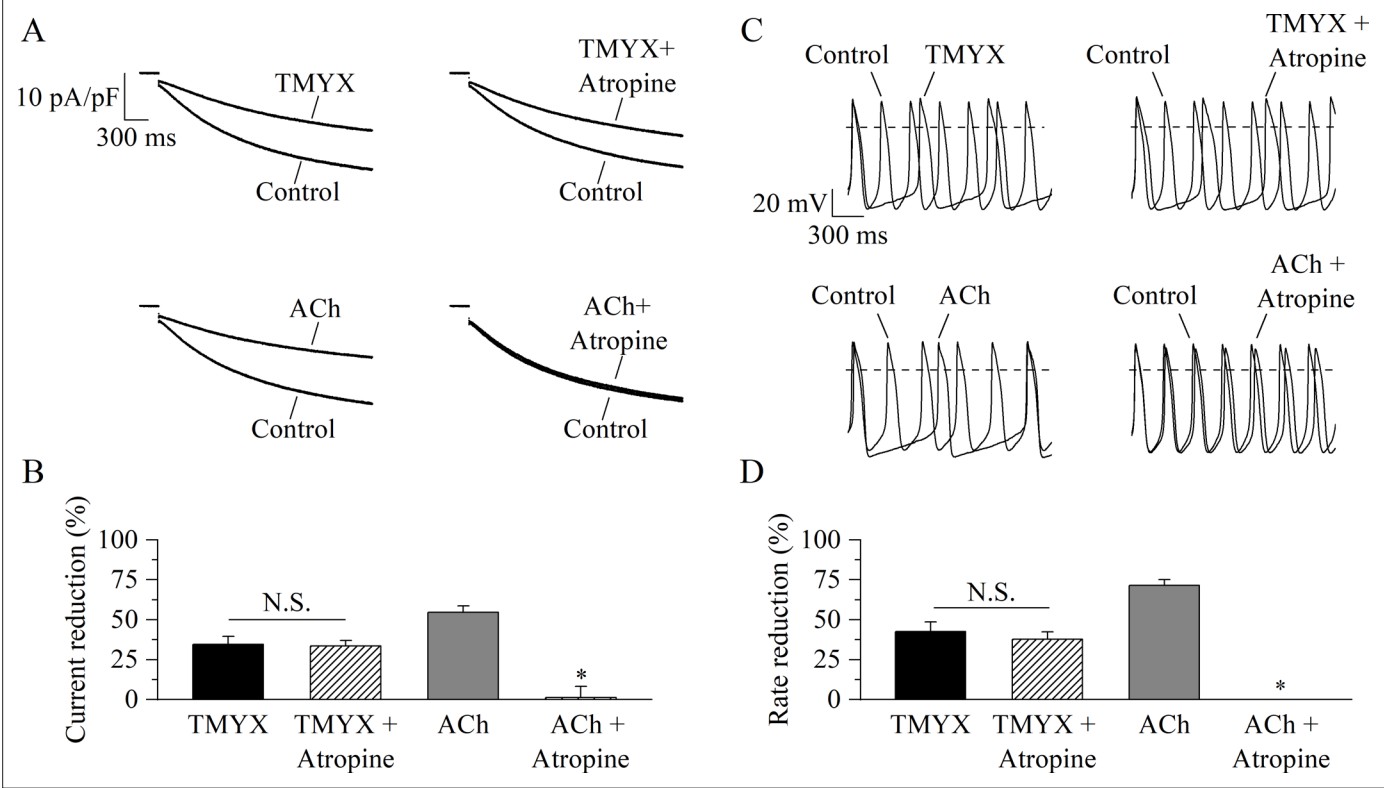

**Figure 3.** Tongmai Yangxin (TMYX) action does not involve muscarinic receptor activation. (**A**) Representative sample current traces recorded during steps to –65 mV in the presence and in the absence of TMYX (6 mg/ml, top) and acetylcholine (ACh) (1 μM, bottom) delivered alone (left) or in combination with atropine (10 μM, right). (**B**) Mean ± SEM steady-state current reduction. Atropine did not modify the action of TMYX (TMYX, –34.6 ± 4.9%; TMYX + atropine, –33.5 ± 3.4%, n = 6) but abolished the effect of ACh (ACh, –54.7 ± 4.0%; ACh+ atropine, 1.2% ± 6.9%, n = 6). N.S. Not significant, p = 0.594; *p < 0.01 vs. ACh (Student's paired t-test). (**C**) Representative action potentials (APs) recorded in the same condition as in panel A. (**D**) Mean ± SEM rate reduction. Atropine did not reduce the ability of TMYX to induce cell bradycardia (TMYX, –42.6 ± 5.8%; TMYX + atropine, –37.6 ± 4.8%, n = 6), but abolished the action of ACh (71.5 ± 3.6%) (n = 7). N.S. Not significant, p = 0.807; *p < 0.01 vs. ACh (Student's paired t-test). Data related to this figure are available in *Figure 3—source data 1*.

The online version of this article includes the following source data and figure supplement(s) for figure 3:

**Source data 1.** Quantification and statistics of TMYX (6 mg/ml) effect on rate and on the funny current amplitude in the presence and in the absence of muscarinic block.

**Figure supplement 1.** The action of Tongmai Yangxin (TMYX) is not mediated by the activation of the adenosine receptor.

**Figure supplement 1—source data 1.** Quantification and statistics of TMYX (6 mg/ml) effect on the funny current in the presence and in the absence of the adenosine receptor block.

**Figure supplement 2.** The action of Tongmai Yangxin (TMYX) is not mediated by PKA activation.

**Figure supplement 2—source data 1.** Quantification and statistics of TMYX (2 mg/ml) effect on APs in the presence of the PKA blocker H-89.

Representative time-courses (left) and current traces (right), recorded in the three different experimental conditions during repetitive hyperpolarizing steps to –65 mV in the absence and presence of TMYX, are presented in *Figure 4A*. A progressive loss of modulatory efficacy of TMYX clearly appears as the intracellular cAMP content increases. As shown in the bar-graph plots in *Figure 4B* a quantitative evaluation of the results yielded the following TMYX-induced current reductions (mean ± SEM): basal cAMP, –49.1 ± 3.0%, n = 15; cAMP 10 μM, –37.9 ± 5.1%, n = 6; cAMP 100 μM, –2.3 ± 3.3% n = 6 (all conditions are significantly different, see legend). The modulatory efficacy of the drug, and its dependence upon intracellular cAMP, was also estimated by means of the ΔV method (empty squares in *Figure 4A*; Material and methods for details) since this analysis allows to assess the shift of the I$_f$ activation curve (*Accili and DiFrancesco, 1996*; *DiFrancesco et al., 1989*). Mean ± SEM TMYX-induced hyperpolarizing ΔV (shift) values were: basal cAMP, 6.3 ± 0.3 mV, n = 15; cAMP 10 μM, 4.5 ± 0.7 mV, n = 6; cAMP 100 μM, 0.45 ± 0.45 mV n = 6 (all significantly different, p < 0.01, one-way

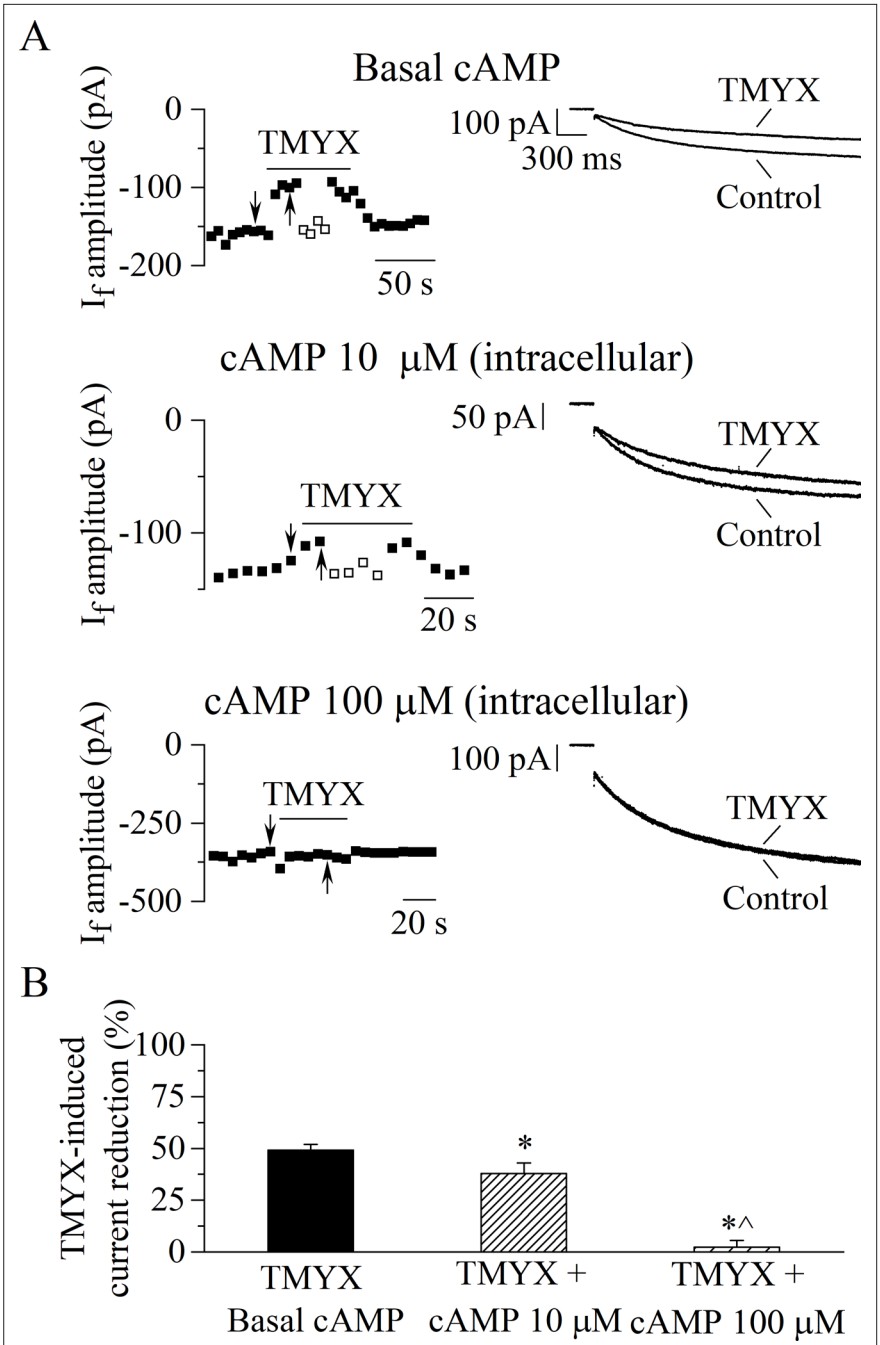

**Figure 4.** The Tongmai Yangxin (TMYX)-induced reduction of the whole-cell I$_f$ is counteracted by increasing concentrations of intracellular cAMP. (**A**) Representative time-courses (left) of the steady-state amplitudes of the current recorded at –65 mV/2.75 s (sample traces, right) in control conditions and in the presence of TMYX (6 mg/ml). Experiments were carried out in the absence (top, n = 15), and in the presence of 10 µM (middle, n = 6) and 100 µM (bottom, n = 6) cAMP in the pipette intracellular solution. Empty squares in top and middle panels indicate steady-state currents recorded after manual adjustment of the holding level (ΔV) to compensate for the inhibitory effect induced by TMYX. (**B**) Bar-graph of the TMYX-induced current reductions (mean ± SEM%) obtained in the three different conditions: TMYX/basal cAMP, –49.1 ± 3.0%, n = 15; TMYX + cAMP 10 µM, –37.9 ± 5.1%, n = 6; TMYX + cAMP 100 µM, –2.3 ± 3.3%, n = 6. *p = 0.049 cAMP 10 µM vs. basal cAMP; *p < 0.01 cAMP 100 µM vs. basal cAMP; ^p < 0.01 cAMP 100 µM vs. cAMP 10 µM (one-way ANOVA followed by Fisher's LSD post hoc test multiple comparisons). Data related to this figure are available in *Figure 4—source data 1*.

The online version of this article includes the following source data and figure supplement(s) for figure 4:

*Figure 4 continued on next page*

*Figure 4 continued*

**Source data 1.** Quantification and statistics of TMYX (6 mg/ml) effects on the funny current in the presence of different cAMP concentrations.

**Figure supplement 1.** Increasing concentrations of intracellular cAMP reduce the $I_f$ inhibitory action of Tongmai Yangxin (TMYX).

**Figure supplement 1—source data 1.** Quantification and statistics of TMYX (2 mg/ml) effects on the funny current in the presence of increasing cAMP concentrations.

ANOVA followed by Fisher's LSD post hoc test multiple comparisons). However, since, in addition to its effect on the activation curve, TMYX also affects the maximal conductance of the current, the ΔV values measured in the experimental paradigm of *Figure 4* represent an underestimation of the absolute shift.

The evidence that the modulatory efficacy of TMYX is counteracted by increasing concentrations of cAMP suggests the intriguing hypothesis of an antagonistic action between these two compounds; additional evidence supporting a mutual interference is presented in *Figure 4—figure supplement 1*. In this case, the cAMP content of SAN cells was experimentally raised by: (i) inhibiting its degradation using a phosphodiesterase (PDE) inhibitor (IBMX, 100 µM) and (ii) favoring its overproduction using an activator of the adenylyl cyclase (Forskolin, 100 µM). The ability of TMYX (6 mg/ml) to reduce the $I_f$ current was then quantified in the presence of different combinations of these substances and of cAMP (10 µM). The bar-graphs shown in *Figure 4—figure supplement 1A,B* confirm the presence of an inverse dependence between cAMP levels and TMYX efficacy. However, these experiments do not provide details on the underlying mechanism.

We therefore proceeded by taking advantage of the inside-out macropatch configuration since this experimental approach allows testing whether TMYX has a membrane-delimited effect (by directly acting on f-channels) or requires instead the involvement of cytoplasmic elements controlling cAMP production and degradation. In *Figure 5—figure supplement 1A*, the $I_f$ current was elicited by a train of hyperpolarizing steps to −105 mV to test the effect of TMYX (6 mg/ml) delivered in the absence of cAMP; no modulation of the current was ever observed (n = 4 patches, p = 0.125, Student's paired t-test). An analysis extended to a wider range of voltages is presented in *Figure 5—figure supplement 1B, C* where a slowly activating voltage ramp (−35/−145 mV) was employed to measure the steady-state I/V curves (panel B) and the associated conductance/voltage (g/V) curve (panel C), in the absence (control) and in the presence of TMYX (6 mg/ml). Statistical analysis revealed that neither half-activation ($V_{1/2}$) nor maximal conductance ($g_{max}$) parameters were affected by the presence of TMYX (*Figure 5—figure supplement 1D*). This evidence demonstrates that TMYX does not influence the intrinsic properties of f-channels and raises the possibility that its inhibitory effect can only occur in the presence of a concurrent cAMP-dependent modulation of the channels.

To verify this possibility, we first evaluated the shift of the $I_f$ voltage dependence induced by cAMP using the ΔV method (*Accili and DiFrancesco, 1996*; *DiFrancesco et al., 1989*) and then the ability of TMYX to reverse this shift (*Figure 5*). Inside-out $I_f$ currents were elicited by a train of hyperpolarizing steps (−105 mV) while membrane patches were exposed to different cAMP concentrations (1, 10, 100 µM) delivered alone and in the presence of TMYX (6 mg/ml). Representative time-courses, current traces, and the corresponding analysis are shown in *Figure 5*. Exposure to cAMP elicited a dose-dependent increase of the current, which was quantified as the voltage correction necessary to restore steady-state control current levels ($ΔV_{cAMP/Cont}$: 6.1 ± 0.5, 12.8 ± 0.5, and 13.6 ± 0.9 mV for 1, 10, 100 µM cAMP, respectively; *Figure 5A*, empty triangles). Addition of TMYX (cAMP+ TMYX) resulted in a reversible reduction of cAMP action quantified as the ΔV correction required to compensate for the effect of TMYX ($ΔV_{TMYX/cAMP}$: 3.5 ± 0.4, 4.7 ± 0.6, and 0 mV for 1, 10, 100 µM cAMP, respectively; *Figure 5A*, empty squares). The difference between experimental $ΔV_{cAMP/Cont}$ and $ΔV_{TMYX/cAMP}$ values represents the cAMP-induced shift in the presence of TMYX ($ΔV_{(cAMP+TMYX)/Cont}$). Dose-dependent $ΔV_{cAMP/Cont}$ (empty triangles) and $ΔV_{(cAMP+TMYX)/Cont}$ (empty circles) values calculated for each patch are plotted in the left panel of *Figure 5B*, and Hill fittings of data points yielded half-maximal concentrations (k) of 1.17 and 5.66 µM for the two conditions, respectively. To better illustrate the antagonism exerted by TMYX (6 mg/ml) on cAMP, we calculated the TMYX-induced fractional inhibition by normalizing the TMYX-induced inhibition of cAMP action ($ΔV_{TMYX/cAMP}$) to the corresponding full cAMP modulation ($ΔV_{cAMP/Cont}$). This procedure was applied both on experimental data points and on the corresponding

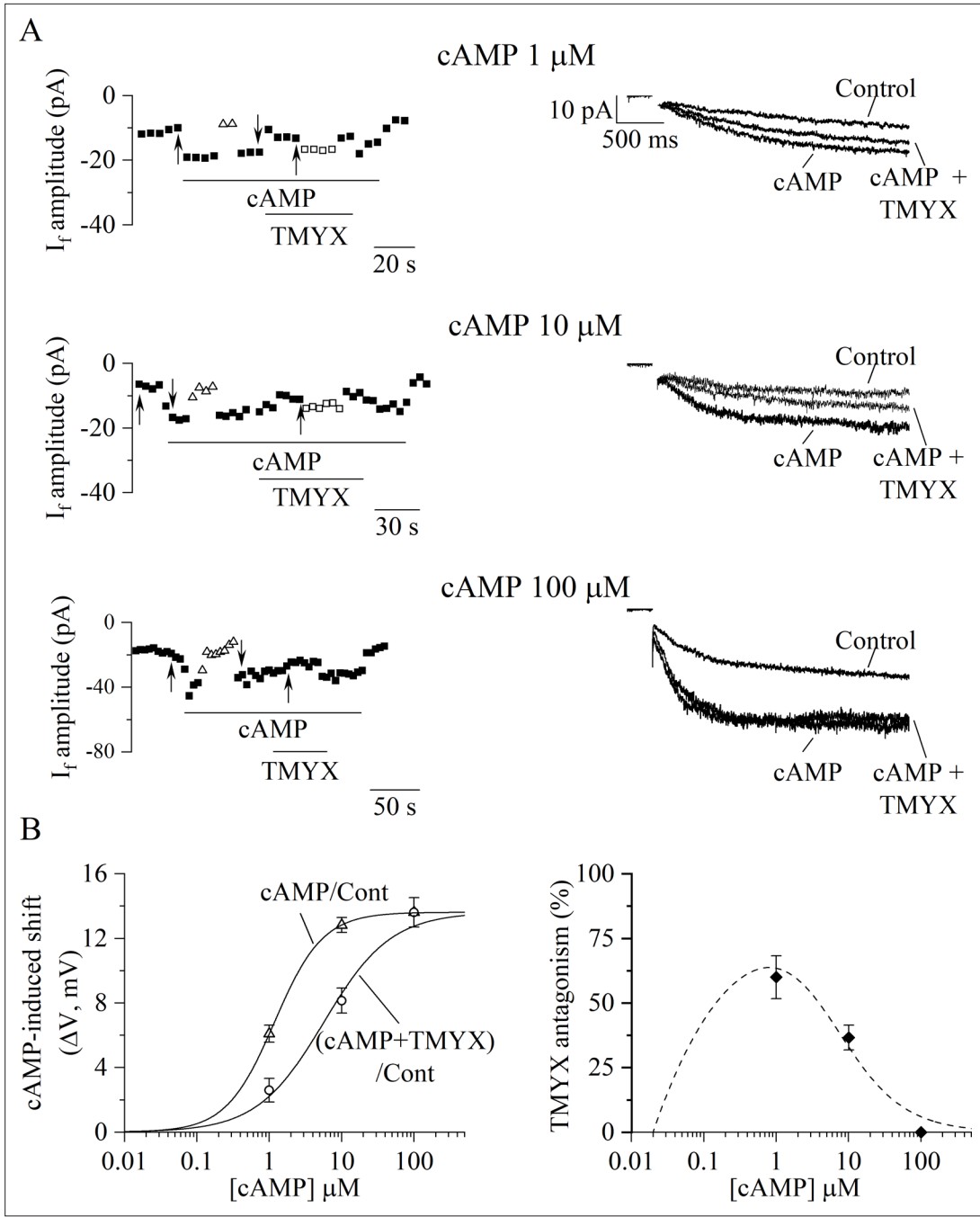

**Figure 5.** Tongmai Yangxin (TMYX) reduces the I_f current by antagonizing its cAMP-induced modulation. (**A**) Sample time-courses (left) and current traces (right) of I_f amplitudes recorded in inside-out macropatches during hyperpolarizing steps (–105 mV); cAMP (1, 10, 100 µM, n = 4–6) was perfused alone (cAMP) or in combination with a fixed dose of TMYX (6 mg/ml, cAMP + TMYX). Empty triangles and squares represent current amplitudes observed after correcting the applied voltage (ΔV) to compensate for (and evaluate) both the effect of cAMP (triangles) and the ability of TMYX to reduce cAMP modulation (squares). (**B**) Left: cAMP-induced shifts of the I_f activation curve obtained in the presence of cAMP alone (cAMP/Cont, empty triangles) and of cAMP+ TMYX ((cAMP+ TMYX)/Cont, empty circles). The continuous lines represent dose-response Hill fittings of experimental data points (cAMP/Cont: k = 1.17 µM and h = 1.30; (cAMP+ TMYX)/Cont: k = 5.66 µM and h = 0.94; 13.6 mV was the maximal shift experimentally measured and was therefore taken as $y_{max}$ for both conditions). (**B**) Right: The antagonism exerted by TMYX on cAMP action was calculated as the fractional inhibition of the I_f current as derived from the Hill fittings (Hill_cAMP/Cont−Hill_(cAMP+TMYX)/Cont)/Hill_cAMP/Cont and from experimental points (diamond symbols, see text for details). Data related to this figure are available in *Figure 5—source data 1*.

The online version of this article includes the following source data and figure supplement(s) for figure 5:

**Source data 1.** Quantification of the TMYX (6 mg/ml) ability to antagonize the cAMP-induced (dose-response) modulation of the funny current.

*Figure 5 continued on next page*

*Figure 5 continued*

**Figure supplement 1.** Tongmai Yangxin (TMYX) does not directly influence the intrinsic properties of f-channels.

**Figure supplement 1—source data 1.** Quantification and statistics of TMYX (6 mg/ml) effects on the intrinsic voltage dependence and maximal conductance of the funny current.

Hill fittings shown in *Figure 5B*, left and results are plotted in *Figure 5B*, right (filled diamonds and dashed line). This distribution demonstrates that in inside-out conditions TMYX antagonizes the effect of cAMP at intermediate (1, 10 µM), but not at high cAMP doses, and the maximal antagonistic effect (63.7%) was observed at a cAMP concentration of 0.8 µM.

## Discussion

Natural botanical compounds commonly used in TCM have recently become of interest also to modern pharmacological studies whose approach is to scientifically challenge their efficacy and to isolate active molecules that could represent novel acquisitions to the western pharmacopeia (*Chen et al., 2006*; *Tang and Huang, 2013*; *Tu, 2016*). Indeed, several studies have demonstrated the safe and beneficial effects of TCM drugs on different pathologies including cancer and cardiovascular diseases (*Efferth et al., 2007*; *Li et al., 2013*; *Pommier, 2006*). Interestingly, cardiovascular TCM drugs often target ion channels; for example, the antiarrhythmic agent Wenxin Keli binds to atrial Na$^+$ channels according to a mechanism of potential relevance in the treatment of atrial fibrillation (*Hu et al., 2016*).

In this study we have characterized the effects of TMYX on the properties of pacemaker cells since this drug is used in TCM to treat cardiovascular diseases including cardiac arrhythmias, coronary artery disease (CAD), and angina (*Cai et al., 2018*; *Fan et al., 2016*).

### At low doses TMYX mainly controls the EDD and the $I_f$ current

Our study indicates a reversible and dose-dependent depression of SAN cell rate due to a robust reduction of the slope of the EDD and by a limited prolongation of the APD (*Figure 1*). These actions are similar to those elicited by the selective $I_f$ blocker ivabradine, which is the only pure heart rate-reducing drug used in western medicine for the treatment of angina and heart failure (*Borer et al., 2012*; *Bucchi et al., 2002*; *Tardif et al., 2013*; *Thollon et al., 1997*). When tested in SAN cells, TMYX (2 mg/ml, *Figure 1B*) and ivabradine (3 µM, *Bucchi et al., 2007*) slow cell rate by 20.8% and 16.2% and prolong the APD by 8.6% and 9.4%, respectively. Similar effects of ivabradine (3 µM) have also been reported in SAN tissue preparation (rate: −19.6/–23.8%, APD50: + 6.7/ + 8.9%, *Thollon et al., 1997*; *Thollon et al., 1994*). Since the APD prolongation represents a pro-arrhythmic effect, the evidence that, at least in single SAN cells, ivabradine and TMYX act similarly on this parameter, suggests a dose-dependent safety of the drug in relation to AP prolongation-dependent arrhythmias. This observation correlates with the use of this drug in TCM clinic.

TMYX exerts a dual action on $I_f$: a negative shift of the voltage dependence and an increase of the maximal conductance, and the former action prevails at physiological voltages (*Figure 2*, *Figure 2—figure supplement 1*, *Figure 2—figure supplement 2*). Interestingly, a negative shift of the activation curve is also the main mechanism during a moderate muscarinic stimulation (*DiFrancesco et al., 1989*) however, this mechanism is not shared with ivabradine. In SAN cells, a moderate cholinergic activation causes a reduction of cell cAMP content, and this associates with a decreased cAMP-dependent modulation of sinoatrial HCN/funny channels; the opposite sequence of events occurs during adrenergic modulation of pacemaker rate (*DiFrancesco, 1993*). For this reason, the shift of the $I_f$ voltage dependence can be considered a readout parameter of the functional interaction between cAMP and HCN/funny channels. cAMP synthesis is operated by the Ca$^{2+}$-sensitive and Ca$^{2+}$-insensitive adenylyl cyclase (AC1/8 and AC5/6, respectively), while cAMP conversion to AMP is catalyzed by the action of the PDE. AC and PDE are therefore central elements of a regulatory pathway that controls cell cAMP dynamics at rest and during autonomic stimulation (*Mika and Fischmeister, 2021*; *Robinson et al., 2021*; *Sirenko et al., 2021*; *Yaniv et al., 2015*). According to *St Clair et al., 2017*, PDE4 is particularly relevant in basal conditions, while PDE3 activity is important during β-adrenergic stimulation. A further level of physiological refinement is provided by the evidence that pacemaker channels are localized in caveolar structures (*Barbuti et al., 2004*; *Barbuti et al., 2012*), and this compartmentalization ensures the existence of functional microdomains where cAMP oscillations may differ from those occurring in

the bulk of the cytoplasm (*Mika and Fischmeister, 2021*). Taken together these regulatory pathways control cAMP levels, hence SAN rate, in pacemaker cells.

## TMYX exerts a direct competitive antagonism on the cAMP-induced activation of the I$_f$ current

Whole-cell experiments presented in *Figure 2B* reveal that the shift induced by TMYX (6 mg/ml) is –11.9 mV, a value similar to the maximal shift induced by ACh (1 μM, 12.6 mV, *Accili et al., 1997*). This comparison thus suggests that the voltage-dependent modulation of I$_f$ induced by 6 mg/ml TMYX should approximate saturation. However, despite a similar effect on the current, 1 μM ACh blocks the spontaneous activity of SAN cells (*DiFrancesco et al., 1989*), while 6 mg/ml TMYX reduces rate only by ~50% (*Figure 1*). Since TMYX does not act on the muscarinic receptor (*Figure 3*), this differ-ence likely arises from the robust cholinergic activation of IK(ACh). While TMYX does not activate the muscarinic receptor, it is conceivable that it may interfere with the cAMP-dependent modulation of the I$_f$ current somewhere along the pathway downstream the receptor. This conclusion is further supported by the evidence that TMYX efficacy is independent from the stimulation of the adenosine receptor and basal PKA activation (*Figure 3—figure supplement 1*, *Figure 3—figure supplement 2*), whose effect on the I$_f$ current is also mediated by a reduction of the cellular cAMP (*Zaza et al., 1996*).

The observation that the inhibitory action of TMYX on the whole-cell I$_f$ is counteracted by increasing concentrations of intracellular cAMP (*Figure 4*, *Figure 4—figure supplement 1*) suggested a func-tional competitive antagonism. This hypothesis was further corroborated by the inside-out exper-iments (*Figure 5*, *Figure 5—figure supplement 1*), which revealed that TMYX does not act in the absence of cAMP. Several studies have shown that cAMP binding to the C-terminus of HCN channels initiates domino-like structural rearrangements leading to the removal of the auto-inhibitory condi-tion which is a hallmark of the cAMP-unbound HCN channels (*Wainger et al., 2001*). The functional aspect of these events is an allosteric-driven shift of the open-close equilibrium toward the open state (*Akimoto et al., 2018*; *DiFrancesco, 1999*; *Wainger et al., 2001*). Our data indicate that TMYX exerts its competitive antagonism either by reducing the channel affinity for cAMP or by interrupting the structural relaxation. The competitive antagonism is clearly illustrated in *Figure 5B*, left where the comparison of the ΔV$_{cAMP/Cont}$ and ΔV$_{(cAMP+TMYX)/Cont}$ dose-response curves displays the hallmarks of allosteric inhibition according to the concerted-symmetry model (*Segel, 1975*): a similar saturating effect (y$_{max}$), a decrease in the half-maximal shifts (k), and a lower Hill coefficient (h). Furthermore, a dissociation constant (k$_i$) value of 1.56 mg/ml was obtained for TMYX by applying the Schild equa-tion (k$_{(cAMP+TMYX)/Cont}$/k$_{cAMP/Cont}$=1+[TMYX]/k$_i$: where k$_{(cAMP+TMYX)/Cont}$ and k$_{cAMP/Cont}$ are half-maximal cAMP-induced shifts in the presence/absence of 6 mg/ml TMYX). This dose is compatible both with the half-inhibitory value observed for the TMYX action on rate (*Figure 1B*) and with the nearly maximal effect on I$_f$ reported for the dose of 6 mg/ml (*Figure 2B* and previous comments).

Interestingly, the neuronal accessory protein TRIP8b modulates the cAMP dependence of HCN channels with a mechanism similar to that of TMYX. For this reason, TRIP8b has raised interest since it may represent a therapeutic target for major depressive disorders (*Hu et al., 2013*; *Lyman et al., 2017*; *Saponaro et al., 2014*).

An important parallelism exists between the actions of β-blockers and TMYX since both reduce the cAMP-induced activation of f-channels: TMYX antagonizes the action of cAMP directly at the channel level (*Figure 5* and *Figure 5—figure supplement 1*), while β-blockers inhibit the β-receptor-cascade and the associated cAMP synthesis.

Despite these different mechanisms, the common functional outcome is the modulation of the I$_f$ activation curve (*Figures 2 and 5*, and *Figure 2—figure supplement 2*). According to the mechanism of action identified in our study, the putative active molecule of TMYX directly regulates the pace-maker f-channel but does not modulate the overall cAMP content of the cell and, for this reason, it is expected to have a selective action on chronotropic control of rate without affecting the inotropism. Multiple effects are instead associated with β-block since a reduction of cell cAMP necessarily affects other processes such as the PKA modulation of other ion channels. For this reason, the identifica-tion of novel pure bradycardic agents is an important pharmacological aim (*Nikolovska Vukadinović et al., 2017*).

Although robust experimental data on the effects of the therapeutic use of TMYX are not yet available, the mechanism of action is compatible with its use in the treatment of CAD and irregular

heartbeat (*Fan et al., 2016*). In the inside-out configuration, the maximal antagonistic effect (63.7%) of TMYX at the dose of 6 mg/ml is observed at a cAMP concentration of 0.8 µM and progressively decreases at higher doses (*Figure 5B*, right). This behavior reveals an additional well-suited physiological and pharmacological feature since it allows recruiting full $I_f$ current and rate modulation when tachycardic stimuli (cAMP levels) are boosted to the maximum. The rationale behind the clinical use of TMYX in TCM is further supported by the evidence that a synthetic derivative of TRIP8b can prevent the β-adrenergic control of SAN cell rate and of $I_f$ (*Saponaro et al., 2018*) and this effect represents proof of principle for further studies and development in applied pharmacology (*Proenza, 2018*).

Finally, it should be mentioned that TMYX is also used to treat premature ventricular complexes (PVCs; personal communication to MB) and, in some cases, PVCs are associated with adrenergic stimuli, high cAMP cell content, and expression of f-channels (*Cantillon, 2013*; *Lee et al., 2019*; *Oshita et al., 2015*). The evidence that ivabradine may prevent this ectopic activity (*Kuwabara et al., 2013*) further supports a causative association and allows to speculate that TMYX could in principle have a therapeutic role.

In conclusion, TMYX slows the spontaneous rate of SAN cells and the underlying mechanism is a selective depression of the diastolic depolarization operated through an antagonistic action on cAMP-induced pacemaker channel activation. Comparison with other pharmacological chronotropic modulators (ivabradine and β-blockers) reveals that TMYX may have an interesting and safe profile. In addition, as pointed out by *Akimoto et al., 2018*, targeting the cAMP binding domain may represent an interesting future perspective for selective modulation of HCN channels since it will reduce the possibility of unspecific interference with other channels. Although TMYX is composed by several components, its mechanism is compatible with the action of a single molecule; we therefore believe that future investigations should focus on this search in addition to provide exhaustive clinical data on TCM patients.

# Materials and methods

## Key resources table

| Reagent type (species) or resource | Designation | Source or reference | Identifiers | Additional information |
|---|---|---|---|---|
| Strain, strain background (*Oryctolagus cuniculus*) | New Zealand rabbit | Charles River | | Female, 35–41 days |
| Chemical compound, drug | Acetylcholine chloride (ACh) | Sigma-Aldrich (Merck) | A6625 | |
| Chemical compound, drug | Adenosine 3′,5′-cyclic monophosphate sodium salt monohydrate (cAMP) | Sigma-Aldrich (Merck) | A6885 | |
| Chemical compound, drug | Forskolin | Sigma-Aldrich (Merck) | F6886 | |
| Chemical compound, drug | 1,3-Dipropyl-8-cyclopentylxanthine (DPCPX) | Sigma-Aldrich (Merck) | C101 | |
| Chemical compound, drug | 3-Isobutyl-1-methylxanthine (IBMX) | Sigma-Aldrich (Merck) | I5879 | |
| Chemical compound, drug | H-89 dihydrochloride hydrate (H-89) | Sigma-Aldrich (Merck) | B1427 | |
| Software, algorithm | pClamp – Clampfit | Molecular Devices | RRID:SCR_011323 | |
| Software, algorithm | pClamp – Clampex | Molecular Devices | RRID:SCR_011323 | Version 10.7 |
| Software, algorithm | Origin | Origin Lab | RRID:SCR_014212 | OriginPro 2020 |
| Software, algorithm | Prism | GraphPad Software | RRID:SCR_002798 | Version 5 |
| Other | Tongmai Yangxin | Le Ren Tang Pharmaceutical Factory | | |

## Animal procedures and cell isolation

All animal procedures performed in this study were carried out in accordance with the guidelines of the care and use of laboratory animals established by the Italian and UE laws (D. Lgs n° 2014/26, 2010/63/UE); the experimental protocols were approved by the Animal Welfare Committee of the Università degli Studi di Milano and by the Italian Ministry of Health (protocol number 1127-2015).

New Zealand female rabbits (0.8–1.2 kg) were anesthetized by intramuscular injection of xilazine (5 mg/kg) and euthanized by an overdose i.v. injection of sodium thiopental (60 mg/kg). The hearts were then quickly removed and placed in pre-warmed (37°C) normal Tyrode's solution (mM: NaCl,

140; KCl, 5.4; CaCl$_2$, 1.8; MgCl$_2$, 1; D-glucose, 5.5; Hepes-NaOH, 5; pH 7.4) containing heparin (10 U/ml). After surgical isolation, the SAN was cut into five to six pieces and treated according to a standard procedure to obtain isolated SAN cells (*Bucchi et al., 2007*). Cells were kept alive and in optimal conditions at 4°C and used for electrophysiological recordings within 48 hr.

## Experimental solutions

Spontaneous APs were recorded from single cells or small beating aggregates; during these recordings the cells were perfused with a normal Tyrode's solution and the patch pipettes were filled with (mM): NaCl, 10; K-aspartate, 130; ATP (Na-salt), 2; MgCl$_2$, 2; CaCl$_2$, 2; EGTA-KOH, 5; Hepes-KOH, 10; creatine phosphate, 5; GTP (Na-salt), 0.1; pH 7.2. Similar solutions were used to record the I$_f$ current in whole-cell condition with the addition of BaCl$_2$ (1 mM) and MnCl$_2$ (2 mM) to the extracellular Tyrode's to block contaminating K$^+$ and Ca$^{2+}$ currents. In inside-out recordings the control solution used to perfuse the intracellular side of the excised patches contained (mM): NaCl, 10; K-aspartate, 130; CaCl$_2$, 2; EGTA-KOH, 5; Hepes-KOH, 10; pH 7.2, and the patch-pipette solution contained (mM): NaCl, 70; KCl, 70; CaCl$_2$, 1.8; MgCl$_2$, 1; BaCl$_2$, 1; MnCl$_2$, 2; Hepes-NaOH, 5; pH 7.4. The resistance of patch pipettes used in whole-cell experiments measured 3–5 MΩ; larger pipettes (0.5–2 MΩ) were used during inside-out macropatch recordings.

TMYX was kindly provided by Le Ren Tang Pharmaceutical Factory (Tianjin, PR China) as a dry powder and is composed of 11 elements, 9 of which are medicinal herbs: radix *Rehmannia glutinosa (15%),* radix and rhizoma *Glycyrrhiza uralensis* (9%, licorice), radix *Ophiopogon japonicus* (9%, dwarf lilyturf), radix *Polygonum Multiflorum* (9%), radix *Codonopsis pilosula* (9%, poor man's ginseng), fructus *Schisandra chinensis* (9%, Chinese magnolia-vine), dried fructus *Ziziphus jujuba* (6%, red date, Chinese date, Chinese jujube), ramulus *Cinnamomum cassia* (3%, Chinese cinnamon), stem of *Spatholobus suberectus* (15%, chicken blood vines), corii colla asini (9%, ejiao, donkey hide gelatin), carapax et plastrum testudinis (6%) (*Cai et al., 2018*; *Fan et al., 2021*; *Fan et al., 2016*).

A stock solution was daily prepared by dissolving the appropriate amount of substance in water (~80°C for 15 min); this solution was then filtered (pore size, 0.45 μm) to remove undissolved components. The stock solution was used to prepare the test solutions at the desired concentrations. ACh, atropine, cAMP, forskolin, 1,3-dipropyl-8-cyclopentylxanthine (DPCPX), IBMX, and dihydrochloride hydrate (H-89) were purchased from Sigma-Aldrich Corporation and used at the concentrations indicated in the text.

Control and test solutions were delivered to the cells through a fast perfusion system or loaded in the whole-cell pipette solution as indicated in the text.

## Patch-clamp experiments and data analysis

Experiments were carried out using the patch-clamp amplifier Axopatch 200B and the pClamp 10.7 software (Molecular Devices, CA); data were analyzed with Clampfit, OriginPro 2020 (Origin Lab, Northampton, MA), Prism 5 (GraphPad Software, San Diego, CA), and a customized software.

APs were recorded from single cells or small beating aggregates and acquired at a sampling rate of 1–2 kHz. After acquisition, AP traces were digitally smoothed by a 10-point adjacent averaging smoothing procedure and the time-derivative calculated according to a second polynomial, 8-point smoothing differentiating routine. AP traces were then processed with customized software to calculate the following parameters: rate (Hz), MDP (the most negative potential for each AP), TOP (the voltage at which the voltage derivative overtakes a fixed threshold of 0.5 mV/ms), EDD (defined as the mean slope in the first half of the diastolic depolarization), APD (the time interval between TOP and the following MDP); additional details can be found in *Bucchi et al., 2007*.

Experimental dose-response points presented in *Figures 1B and 5B* were fitted to the Hill equation ($y = y_{max}/(1+(k/x)^h)$), where $y_{max}$ is the maximal effect, x is the drug concentration, k is the drug concentration eliciting half-maximal block (*Figure 1B*) or half maximal shift (*Figure 5B*), and h is the Hill factor.

The activation curves of the whole-cell I$_f$ current were obtained by applying a train of two consecutive voltage steps: the first pulse was delivered at test potentials (from –20 to –125 mV, increment between steps: –15 mV) to attain steady-state current activation, while the second step was delivered at –125 mV to ensure maximal activation. Normalized tail currents amplitudes at –125 mV represent the activation variable at each test potential. Mean ± SEM fractional activation values, measured in

control condition and in the presence of different concentrations of TMYX, were interpolated by the Boltzmann distribution (y = 1/(1+ exp((V−$V_{\frac{1}{2}}$)/s))), where y is the fractional activation, V is voltage, $V_{\frac{1}{2}}$ is the half-activation voltage, and s is the inverse-slope factor.

The fully activated I/V relations in whole-cell condition were obtained by applying a voltage protocol consisting of two sequential pulses: the cell was first hyperpolarized to –125 mV and then depolarized to different test potentials in the range –120/+20 mV (increment: 20 mV). After leakage correction the tail currents were normalized and plotted as a function of tail step voltages. Mean ± SEM experimental values were interpolated by a linear fit ($I_{density}$=(a*V + b)) to yield the fully activated (I/V) curves.

The steady-state I/V curves shown in *Figure 2D*, *Figure 2—figure supplement 1D* were obtained by multiplying the Boltzmann fitting of fractional activation and the linear fitting of the fully activated I/V curve.

During macropatch inside-out experiments shown in *Figure 5* the $I_f$ current amplitude was elicited by hyperpolarizing steps to –105 mV from a holding potential of –35 mV. Steady-state I-V relations (*Figure 5—figure supplement 1B*) were recorded by means of hyperpolarizing ramps from –35 to –145 mV at a rate of –110 mV/min. Inside-out activation curves (*Figure 5—figure supplement 1C*) were derived from the steady-state I-V currents (see *Bois et al., 1997* for details) and fitted to the Boltzmann equation.

In some analyses (*Figures 4 and 5*) the effect of TMYX on the $I_f$ current was assessed by means of the ΔV method. This method quantifies the manual voltage adjustment (ΔV) of the holding potential which is introduced during drug delivery to compensate for the drug-induced current reduction and restore a steady-state current amplitude as the one observed prior to drug delivery (that is in control condition). The ΔV parameter measured in this condition was originally used by several authors (*Accili and DiFrancesco, 1996*; *Altomare et al., 2003*; *Bois et al., 1997*) to evaluate the shift of the activation curve induced by modulatory agents. In whole-cell experiments presented in *Figure 4*, however, this parameter (see text) is an underestimation of the shift of the activation curve since the ΔV adjustment must also compensate for the TMYX-induced increase of channel conductance. This limitation does not apply to data presented in *Figure 5* since in the inside-out configuration (*Figure 5—figure supplement 1B*) we could not measure any increase of the channel conductance.

Whole-cell and inside-out experiments were carried out at 35°C ± 0.5°C and at room temperature, respectively.

## Statistical analysis

No statistical method was used to predetermine sample size, but our samples sizes are similar to those reported in previous studies (*Altomare et al., 2006*; *Bucchi et al., 2007*; *Milanesi et al., 2006*; *Thollon et al., 1994*; *Van Bogaert and Pittoors, 2003*).

All data are presented as mean ± SEM values. Group comparisons were analyzed for statistical significance using Student's paired t-test (correlated samples, *Figures 1C, 3B and D*, and *Figure 3—figure supplement 1B*, *Figure 5—figure supplement 1A*,D) or one-way ANOVA followed by Fisher's LSD post hoc test (multiple comparisons, *Figure 4B* and *Figure 4—figure supplement 1A, B*; multiple comparison for repeated measurements, *Figure 3—figure supplement 2B*).

Activation curves were compared using the extra sum-of-squares F test, while the slopes of I/V relations were evaluated through the linear regression analysis test (*Figure 2B and C* and *Figure 2—figure supplement 1B, C*).

Computational and statistical analysis were carried out with OriginPro 2020, OriginLab, Northampton, MA, and GraphPad Prism 5, GraphPad Software, San Diego, CA.

Statistical significance is indicated by p-values < 0.05. Exact p values are provided except when p < 0.01.

## Additional information

### Competing interests

Dario DiFrancesco: DiFrancesco Dario was supported with a grant the research study on the Chinese medicine drug TMYX by Tianjin Zhongxin Pharmaceutical Group Co., Ltd. Le Ren Tang Pharmaceutical Factory who had absolutely no part in the study plan, data collection and analysis, and manuscript

writing. The money was given to the University of Milan and no Honorarium was ever paid to DiFrancesco Dario. The author has no other competing interests to declare. Annalisa Bucchi: was supported with a grant the research study on the Chinese medicine drug TMYX by Tianjin Zhongxin Pharmaceutical Group Co., Ltd. Le Ren Tang Pharmaceutical Factory who had absolutely no part in the study plan, data collection and analysis, and manuscript writing. The money was given to the University of Milan and no Honorarium was ever paid to Annalisa Bucchi. The author has no other competing interests to declare. Mirko Baruscotti: was supported with a grant the research study on the Chinese medicine drug TMYX by Tianjin Zhongxin Pharmaceutical Group Co., Ltd. Le Ren Tang Pharmaceutical Factory who had absolutely no part in the study plan, data collection and analysis, and manuscript writing. The money was given to the University of Milan and no Honorarium was ever paid to Mirko Baruscotti. The author has no other competing interests to declare. The other authors declare that no competing interests exist.

### Funding

| Funder | Grant reference number | Author |
|--------|------------------------|--------|
| Le Ren Tang | CTE_INT18DDIFR_01 | Dario DiFrancesco<br>Annalisa Bucchi<br>Mirko Baruscotti |

The funders had no role in study design, data collection and interpretation, or the decision to submit the work for publication.

### Author contributions

Chiara Piantoni, Manuel Paina, Conceptualization, Data curation, Formal analysis, Investigation, Methodology, Writing – original draft, Writing – review and editing; David Molla, Formal analysis, Investigation, Writing – original draft, Writing – review and editing; Sheng Liu, Conceptualization, Funding acquisition, Methodology, Project administration, Writing – original draft, Writing – review and editing; Giorgia Bertoli, Hongmei Jiang, Formal analysis, Writing – original draft; Yanyan Wang, Investigation, Writing – original draft, Writing – review and editing; Yi Wang, Conceptualization, Writing – review and editing; Yi Wang, Conceptualization, Funding acquisition, Writing – review and editing; Dario DiFrancesco, Conceptualization, Formal analysis, Funding acquisition, Investigation, Validation, Writing – original draft, Writing – review and editing; Andrea Barbuti, Writing – review and editing; Annalisa Bucchi, Mirko Baruscotti, Conceptualization, Formal analysis, Funding acquisition, Investigation, Methodology, Project administration, Supervision, Validation, Writing – original draft, Writing – review and editing

### Author ORCIDs

Chiara Piantoni http://orcid.org/0000-0002-3621-8402
Manuel Paina http://orcid.org/0000-0001-9250-9303
David Molla http://orcid.org/0000-0003-4355-4508
Sheng Liu http://orcid.org/0000-0001-8160-5762
Giorgia Bertoli http://orcid.org/0000-0001-5352-253X
Hongmei Jiang http://orcid.org/0000-0001-5117-5212
Yanyan Wang http://orcid.org/0000-0003-4859-4963
Yi Wang http://orcid.org/0000-0002-3676-9183
Yi Wang http://orcid.org/0000-0001-5098-6750
Dario DiFrancesco http://orcid.org/0000-0002-7322-1790
Andrea Barbuti http://orcid.org/0000-0002-4521-4913
Annalisa Bucchi http://orcid.org/0000-0002-5303-4242
Mirko Baruscotti http://orcid.org/0000-0002-6155-8388

### Ethics

All animal procedures performed in this study were carried out in accordance with the guidelines of the care and use of laboratory animals established by the Italian and UE laws (D. Lgs n° 2014/26, 2010/63/UE); the experimental protocols were approved by the Animal Welfare Committee of the Universita degli Studi di Milano and by the Italian Ministry of Health (protocol number 1127-2015).

### Decision letter and Author response

Decision letter https://doi.org/10.7554/eLife.75119.sa1

Author response https://doi.org/10.7554/eLife.75119.sa2

## Additional files

### Supplementary files
• Transparent reporting form

### Data availability
All raw data generated and analyzed during this study have been uploaded as source data files.

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
