## [Editor Report]

Tongmai Yangxin (TMYX) is a complex compound of Traditional Chinese Medicine used to treat several cardiac rhythm disorders; however, no information regarding its mechanism of action is available. This study provides mechanistic insight into where TMYX acts to inhibit the pacemaking current called If. In some respects TMYX behaves like a β blocker or muscarinic antagonist, but it works to inhibit the ion channel to maximum effect when cAMP concentrations are low, thus allowing the full effect of sympathetic stimulation to still occur on I_f_ when metabolic rate is high. This compound therefore has the potential for high therapeutic utility to control cardiac arrhythmia.

---

## [Decision Letter]

**Decision letter after peer review:**

Thank you for submitting your article "The natural compound TMYX reduces SAN cells rate by antagonizing the cAMP modulation of f-channels" for consideration by *eLife*. Your article has been reviewed by 3 peer reviewers, one of whom is a member of our Board of Reviewing Editors, and the evaluation has been overseen by Richard Aldrich as the Senior Editor. The following individual involved in review of your submission has agreed to reveal their identity: Denis Noble (Reviewer #3).

Essential revisions:

1) Change title. Make broader. eg 'Natural compound reduces cardiac excitability by modulating cAMP dependent pacemaking channel' or 'Chinese natural compound decreases cardiac pacemaking by targeting second messenger regulation of hyperpolarization activated ion channel'

2) Details of TMYX composition.

*Reviewer #1:*

The authors provide a detailed electrophysiological characterisation underpinning the putative antiarrhythmic action of a novel herbal medicine TMYX.

Data supports the idea that this compound targets the second messenger cAMP to reduce activation of the pacemaker current If.

The precise way this is achieved has not been discovered, but the initial observation is of wider interest. Finding the precise chemical ingredients that convey inhibition of I_f_ at the molecular level would be an important next step.

Details of the active ingredients of TMYX would be helpful.

Some discussion about PDEs and their interaction with cAMP is warranted to improve the scholarship of the manuscript. In addition, reference ought to be made microdomain cAMP signalling.

Probing PDE signalling would certainly enhance the manuscript given the main result. I am left wondering whether the action of TMYX is via PDE2 hydrolysis of cAMP to decrease If.

Acknowledgement of its action on decreasing cGMP also worth discussing.

*Reviewer #2:*

Strengths:

1) Provide interesting data to support multi-components TCM medicine for the treatment cardiac arrhythmias.

2) Provide detailed single-cell electrophysiological characterization of TMYX on action potentials of SAN cells by eliciting a reversible and dose-dependent slowing of spontaneous action potentials rate by a selective reduction of the diastolic phase. This action is mediated by a negative shift of the I_f_ activation curve and is caused by a reduction of the cAMP-induced stimulation of pacemaker channels.

3) Provide the evidence that TMYX acts by antagonizing the cAMP-induced modulation of the pacemaker channels.

4) TMYX exerts its maximal antagonistic action at submaximal cAMP concentrations and then progressively becomes less effective thus ensuring a full contribution of I_f_ to pacemaker rate during high metabolic demand and sympathetic stimulation.

Weaknesses:

1) The ionic mechanisms of TMYX on the EDD of the pacemaker cells require investigations on other ion channels contributing to the EDD.

2) Lack of evidence to support cAMP-induced allosteric modulation of the pacemaker channels by TMYX such as identification of the cAMP binding domain and structural conformation changes.

3) The key active components of TMYX responsible for the therapeutic effects observed in this study, for example, the effects of components flavonoids, coumarins, iridoid glycosides, saponins and lignans as authors mentioned in the introduction.

4) Can authors provide the effects of key active components of Tongmai Yangxin (TMYX) on cAMP, if and APs?

5) Does TMYX modulate PKA activity and its downstream targets?

6) What is dose range of TMYX for in vivo treatment?

*Reviewer #3:*

The western scientific world is waking up to the therapeutic utility of multi-component medications developed in East Asia over thousands of years. The therapies differ from most western remedies in combining very many chemicals in mixtures that are thought to act in complementary ways, instead of relying on a single chemical to exert the therapeutic effects. This paper applies rigorous experimental procedures to reveal the cellular processes by which the 5 herbal mixed medication known as TMYX exerts its potentially beneficial slowing of the heart. The article is an excellent example of how East Asian traditional medicine can be understood in western scientific terms and so contributes greatly to medical understanding of an important traditional medication.

The paper is ready for publication.

---

## [Author Response]

Essential revisions:1) Change title. Make broader. eg 'Natural compound reduces cardiac excitability by modulating cAMP dependent pacemaking channel' or 'Chinese natural compound decreases cardiac pacemaking by targeting second messenger regulation of hyperpolarization activated ion channel'

We agree with this suggestion. The new title is “Chinese natural compound decreases cardiac pacemaking of rabbit sinoatrial node cells by targeting second messenger regulation of hyperpolarization activated ion channel”. Page 1, lines 1,2 (in grey)

2) Details of TMYX composition.

We have now provided specific details (latin names and, when possible, common names) and the % of composition of each of the 11 components of TMYX; a novel reference has also been added. This part now appears in the Materials and methods section. Page 18, lines 408-416 (in grey).

Reviewer #1:The authors provide a detailed electrophysiological characterisation underpinning the putative antiarrhythmic action of a novel herbal medicine TMYX.Data supports the idea that this compound targets the second messenger cAMP to reduce activation of the pacemaker current If.The precise way this is achieved has not been discovered, but the initial observation is of wider interest. Finding the precise chemical ingredients that convey inhibition of I_f_ at the molecular level would be an important next step.Details of the active ingredients of TMYX would be helpful.

TMYX is a complex drug composed by more than 80 molecules including flavonoids, coumarins, iridoid glycosides, saponins and lignans (Tao et al., 2015, DOI: 10.1002/jssc.201401481 J). Several studies have demonstrated that TMYX exerts anti-inflammatory and anti-tumor effects and also reduces oxidative stress (Cai et al., 2018, doi: 10.1016/j.jchromb.2018.09.038; Tao et al., 2015; doi: 10.1002/jssc.201401481; Fan et al., 2021, doi: 10.1016/j.jep.2021.114106). Obviously, a match between single active molecules and the corresponding effect represents the aim of the modern approach to TCM, and novel information is starting to emerge. For example, Tao et al., (2015, DOI: 10.1002/jssc.201401481 J) report that six active compounds (gomisin D, schisandrin, glycyrrhizic acid, 2,3,5,4’-tetrahydroxystilbene-2-O-β-D-glucoside, formononetin, and ononin) exert dose-dependent anti-inflammatory activities.

Clinical metabolomic analysis have also reported that TMYX exerts therapeutic effects on stable angina and more in general on arrhythmias (Cai et al., 2018, doi: 10.1016/j.jchromb.2018.09.038). In the effort to identify the molecular biological basis of the observed cardiac improvements these authors carried out a clinical trial that led to the conclusion that “TMYX exerts its effects by improving myocardial energy supply disorder and aminoacid dysfunction and attenuating oxidative stress and inflammation” (their figure 3). TMYX also exerts a protective effect on cisplatin-induced cardiotoxicity by regulating the Nrf2 and p38-MAPK pathways (Cui et al., 2018, doi: 10.1016/j.biopha.2018.09.095).

We believe that our finding adds one important piece of information to this complexity since we conclude that a molecular component of TMYX (or one of its biological metabolites) is able to uncouple the pacemaker f/HCN channels from their cAMP-induced modulation, and this action has the feature of competitive antagonism. As also requested in Point 2 of the “Essential Revisions”, we have now expandend the Material and Methods section to include additional details of each component of TMYX. Page 18, lines 408-416 (in grey).

Some discussion about PDEs and their interaction with cAMP is warranted to improve the scholarship of the manuscript. In addition, reference ought to be made microdomain cAMP signalling.Probing PDE signalling would certainly enhance the manuscript given the main result. I am left wondering whether the action of TMYX is via PDE2 hydrolysis of cAMP to decrease If.Acknowledgement of its action on decreasing cGMP also worth discussing.

We have now extended the discussion on cAMP formation and degradation and in particularly we mention the fundamental role of the Adenylil Cyclases (both the Ca-sensitive and insensitive isoforms) and of the Phosphodiesterases in cAMP production and degradation; consequently, ACs and PDEs indirectly modulate several ion channels. As suggested by the reviewer, it would be important to verify whether TMYX exerts any effect on PDE and on AC, however, we believe that this approach should be thoroughly explored once the active molecule responsible for the effect on the I_f_ current will be isolated. Indeed, we cannot exclude that different molecules may act on these different targets, but our interest is to strictly identify additional and potentially undesired side effects of the molecule specifically targeting the f-channels. Finally, we have now acknowledged that cAMP signaling may also be spatially confined due to the existence of caveolar microdomains. These modifications now appear in the Discussion session. Pages 11/12, lines 289-301(in grey).

Reviewer #2 (Recommendations for the authors):4) Can authors provide the effects of key active components of Tongmai Yangxin (TMYX) on cAMP, if and APs?

TMYX is composed by a large number of single molecules (see Answer 1 to Reviewer 1), however our study does not identify the active principle that modulates the I_f_ current and thus SAN cell rate. We are currently working on this aspect by chemical fractioning of TMYX.

The experiments presented in this paper demonstrate the following points:

– TMYX reduces SAN cell rate mainly by a dose-dependent action on the early diastolic depolarization phase;

– TMYX exerts a dual action on the I_f_ current: increases the maximal conductance and decrease the voltage dependence of channel availability; the latter effect prevails at physiological voltages;

– TMYX does not modulate the intrinsic properties of the channels, as demonstrated by the lack of effect on inside-out condition, but it antagonizes the cAMP-induced activation process. Hence we describe TMYX as a competitive antagonist. Whether this action is exerted at the cAMP binding pocket or elsewhere downstream the functional structural pathways that couples cAMP binding to channel opening is still unclear.

5) Does TMYX modulate PKA activity and its downstream targets?

The inside-out experiments presented in Figure 5 (and Figure 5-suppement figure 1) clearly show that TMYX antagonizes the direct action of cAMP on f-channels. However, we could not exclude that in the wholecell condition the action of TMYYX could be, either directly or indirectly, influenced by the activity of PKA. As stated in the Answer 2 to Reviewer 1 we believe that a complete study on the action of TMYX on other targets (such as PKA) should be thoroughly explored once the active molecule responsible for the effect on the I_f_ current will be isolated. However, since PKA modulates f-currents we followed the reviewer’s suggestion and we tested whether the effect of TMYX on SAN cell rate may also partly depend on an inhibitory action on PKA. We recorded spontaneous action potentials from SAN cells and we compared the effects of TMYX alone and in combination with the PKA inhibitor H-89. We chose to use a concentration of H-89 of 1 µM to avoid the unspecific effects of H-89. Indeed, the product information

(https://www.sigmaaldrich.com/IT/it/product/mm/371963m?gclid=Cj0KCQiA0eOPBhCGARIsAFIwTs6s1Q4e1GdhTbJSTpeJvthqYX3DqPgYsn0uE5Npembinformaitnwybs1JNqHe8oaAnGwEALw_wcB) provides the following Ki values: PKA: 48 nM; CaMKII: 29.7 µM; PKC:31.7 µM, therefore at 1 µM H-89 exert its effects on PKA with minimal or no effects on PKC and CAMKII.

Our experiment clearly demonstrates that the effects of TMYX assessed prior to (-27.5%) and in the presence of H-89 (-27.6%) are not different. These data are now inserted in the novel Figure 3-supplement figure 2 and discussed in the Result and Discussion sections (Pages 7, lines 168-175 and Page 12/13 lines 314-315) (in grey). This experiment excludes that at rest PKA inhibition influences the inhibitory action of TMYX on rate.

6) What is dose range of TMYX for in vivo treatment?

The suggested dose for human in-vivo treatment is generally 4-8 grams qd for 8 weeks. This information was communicated by the Chinese partners and also reported by Fan et al., doi 10.1016/j.jep.2021.114106. This information is now mentioned on Page 3, line 71 (in grey).

Reviewer #3 (Recommendations for the authors):The western scientific world is waking up to the therapeutic utility of multi-component medications developed in East Asia over thousands of years. The therapies differ from most western remedies in combining very many chemicals in mixtures that are thought to act in complementary ways, instead of relying on a single chemical to exert the therapeutic effects. This paper applies rigorous experimental procedures to reveal the cellular processes by which the 5 herbal mixed medication known as TMYX exerts its potentially beneficial slowing of the heart. The article is an excellent example of how East Asian traditional medicine can be understood in western scientific terms and so contributes greatly to medical understanding of an important traditional medication.The paper is ready for publication.

We thank the reviewer for his kind appreciation of our study. We have now modified the Materials and methods section to provide the latin names (and when possible also the common names), the part of the plants from which the extracts are made, and the relative abundance (%) of each of the 11 components of TMYX. This part now appears in the Materials and methods section. Page 18, lines 408-416 (in grey).